# *Spirulina maxima* Derived Pectin Nanoparticles Enhance the Immunomodulation, Stress Tolerance, and Wound Healing in Zebrafish

**DOI:** 10.3390/md18110556

**Published:** 2020-11-07

**Authors:** Dinusha C. Rajapaksha, Shan L. Edirisinghe, Chamilani Nikapitiya, SHS Dananjaya, Hyo-Jung Kwun, Cheol-Hee Kim, Chulhong Oh, Do-Hyung Kang, Mahanama De Zoysa

**Affiliations:** 1College of Veterinary Medicine, Chungnam National University, Yuseong-gu, Daejeon 34134, Korea; dinusharajapaksha@o.cnu.ac.kr (D.C.R.); shanlakmal@o.cnu.ac.kr (S.L.E.); chamilani14@cnu.ac.kr (C.N.); shsdananjaya@cnu.ac.kr (S.D.); hyojung@cnu.ac.kr (H.-J.K.); 2Department of Biology, Chungnam National University, Yuseong-gu, Daejeon 34134, Korea; zebrakim@cnu.ac.kr; 3Jeju Marine Research Center, Korea Institute of Ocean Science and Technology (KIOST), Jeju Special Self-Governing Province 63349, Korea; och0101@kiost.ac.kr; 4Department of Ocean Science, University of Science and Technology (UST), Jeju Special Self-Governing Province 63349, Korea

**Keywords:** *Aeromonas hydrophila*, immunomodulation, pectin nanoparticles, *Spirulina maxima*, stress tolerance, wound healing, zebrafish

## Abstract

In this study, *Spirulina maxima* derived pectin nanoparticles (SmPNPs) were synthesized and multiple biological effects were investigated using in vitro and in vivo models. SmPNPs were not toxic to Raw 264.7 cells and zebrafish embryos up to 1 mg/mL and 200 µg/mL, respectively. SmPNPs upregulated Il 10, Cat, Sod 2, Def 1, Def 2, and Muc 1 in Raw 264.7 cells and *tlr2, tlr4b, tlr5b, il1β, tnfα, cxcl8a, cxcl18b, ccl34a.4, ccl34b.4, muc5.1, muc5.2, muc5.3, hamp, cstd, hsp70, cat*, and *sod1* in the larvae and adult zebrafish, suggesting immunomodulatory activity. Exposure of larvae to SmPNPs followed by challenge with pathogenic bacterium *Aeromonas hydrophila* resulted a two-fold reduction of reactive oxygen species, indicating reduced oxidative stress compared to that in the control group. The cumulative percent survival of larvae exposed to SmPNPs (50 µg/mL) and adults fed diet supplemented with SmPNPs (4%) was 53.3% and 76.7%, respectively. Topical application of SmPNPs on adult zebrafish showed a higher wound healing percentage (48.9%) compared to that in the vehicle treated group (38.8%). Upregulated wound healing markers (*tgfβ1*, *timp2b*, *mmp9*, *tnfα*, *il1β,*
*ccl34a.4*, and *ccl34b.4*), enhanced wound closure, and restored pigmentation indicated wound healing properties of SmPNPs. Overall, results uncover the multiple bioactivities of SmPNPs, which could be a promising biocompatible candidate for broad range of aquatic and human therapies.

## 1. Introduction

Marine compounds from different sources possess a variety of biological activities, such as antimicrobial, antihypertensive, anticancer, antioxidant, and anti-inflammatory [1,2], and are, thus, widely used for pharmaceutical and biomedical applications. Most of these biomaterials can significantly control the immune system through different mechanisms to process effective reactions. However, some synthesized drugs cause side effects and toxicity, which restrict their application [3]. Compared to the synthesized drugs, most of the naturally derived drugs from marine sources, such as algae and micro algae, are gaining popularity owing to their better effects, lesser toxicity, and lower costs [4].

Spirulina is referred to as free-floating filamentous microalga with multiple effects [5]. *Spirulina platensis*, *S. maxima*, and *S. fusiformis* are the most intensively investigated species that are edible, rich in nutrients, and have potential therapeutic value [6]. Different biological activities, such as anti-inflammatory, antiviral, and cholesterol lowering activities of Spirulina are well documented [7]. Pectin is a biopolymer with a linear anionic backbone consisting of α-1,4-linked D-galacturonic acid and α-1,2-l-rhamnose units, numerous neutral sugars, such as galactose, arabinose, and a lesser number of other sugars [4,8]. Currently, pectin has evinced interest among scientists for medicinal and pharmacological applications owing to its high availability, and ability to resist digestive enzymes and maintain macromolecular structural patterns of its sugar chains in the stomach and small intestine [9]. Furthermore, it can regulate immunological surveillance, such as anti-inflammatory, immunomodulatory, antibacterial, hypoglycaemic, and antioxidant activities, and can modulate desirable physiological functions, such as reduction of digestive complications [10], and lowering of the low-density lipoprotein (LDL) and cholesterol levels [11]. Activation of cell death pathways is another property of pectins that plays an important role in preventing certain types of cancer [12]. Previously, we also reported enhanced immunomodulation, disease resistance [13] and wound healing [14] properties of *S. maxima* derived pectin (SmP). However, marine algae are the least documented sources of pectin, which contain large quantities of soluble and insoluble dietary fibers. Physicochemically they differ from land plants, but have similar promising functional properties [15].

Nanomaterials increase the cellular uptake of drugs owing to their physicochemical properties, such as large surface area, mechanical strength, optical activity, and high chemical reactiveness. Hence, they provide uniqueness and suitability for various physicochemical and biological applications [16]. Considering these properties, widespread attention is being paid towards the use of nanomaterials for drug delivery [17]. Several studies have proven the enhanced effects of nano-sized biopolymers. For instance, Nikapitiya et al. [18] suggested chitosan nanoparticles as positive immune response modulators in zebrafish larvae against *Aeromonas hydrophila* infection. Burapapadh et al. [19] developed pectin nanoparticles through mechanical homogenisation and showed enhanced drug dissolution. Moreover, we previously reported that the marine microalga *S. maxima* derived modified pectin (MSmP) and its nanoparticles (MSmPNPs) could modulate the gut microbiota of mice, promote multiplication of desired gut microbial communities, and trigger immune responses, and provided evidence for increased bioavailability [20]. Nevertheless, without any further modifications to SmP, *S. maxima* pectin nanoparticles (SmPNPs) and their therapeutic potentials have not been fully evaluated.

In this study, we investigated the various biological effects (immunomodulation, stress tolerance, disease resistance, and wound healing) of synthesized SmPNPs using in vitro and in vivo models. The immunomodulatory effect was investigated in SmPNPs treated Raw 264.7 murin macrophage cells, zebrafish larvae and in the gut of adult zebrafish upon a SmPNPs (4%) supplemented diet fed for 6 weeks. Moreover, the disease resistance capacity of *A. hydrophila* challenged larvae and adults was investigated. The effect of SmPNPs in reducing oxidative stress in zebrafish larvae infected with *A. hydrophila* was also investigated. Furthermore, the wound healing effect of SmPNPs by topical application to laser wounded zebrafish was investigated, and time course analyses of dermal wound closure, percentage and rate of wound healing, pigment restoration, and transcriptional immune responses were performed.

## 2. Results

### 2.1. Physicochemical Properties and Toxicity of SmPNPs In Vitro and In Vivo

The prepared SmPNPs had irregular shapes and they were soluble in water. The particle size and zeta potential were 125.3 nm and −24.1 mv, respectively (Appendix A). The cytotoxicity of SmPNPs in Raw 264.7 cells was investigated using the 3-(4,5-dimethylthiazol-2-yl)-2,5-diphenyltetrazolium bromide (MTT) assay. SmPNPs concentrations up to 1 mg/mL were found to be non-toxic; however, the viability of cells was reduced at concentrations above 1 mg/mL (Appendix A). The IC_50_ of the SmPNPs was 5.60 mg/mL; therefore, we used 0.5 and 1 mg/mL SmPNPs for further in vitro experiments. The in vivo toxicity was evaluated by determining cumulative mortality and hatching percentage (%) upon exposure of zebrafish embryos (1 h post fertilization [hpf]) to different doses (50, 100, 200, 400, and 600 µg/mL) of SmPNPs. No significant difference (*p* > 0.05) in cumulative mortality, with respect to that in the control, was observed at 96 hpf up to 200 µg/mL (Appendix A). Similarly, no significant differences (*p* > 0.05) were observed in the hatching percentage (%) at 60 hpf between SmPNPs treatment (50 and 100 µg/mL) and the control (Appendix A). Although the mortality in the 200 µg/mL SmPNPs treatment was not significantly different, the hatching percentage was significantly (*p* < 0.05) lower at 60 hpf, when compared to that in the control. Upon exposure to high doses (400 and 600 µg/mL) of SmPNPs, the cumulative mortality was significantly (*p* < 0.05) increased (18 and 60%, respectively) and LD_50_ was 547 µg/mL. At ≥ 200 µg/mL of SmPNPs, the hatching percentage (%) at 60 hpf was significantly (*p* < 0.05) decreased. Moreover, typical malformations including pericardial oedema, yolk sac oedema, and axial and head malformations were observed at high doses (> 200 µg/mL) of SmPNPs at 60 hpf (Appendix A).

### 2.2. Transcriptional Profiling of Immune-Related Genes in Raw 264.7 Cells Exposed to SmPNPs

The results of transcriptional profiling of immune-related genes in Raw 264.7 cells is shown in Figure 1. Pro- and anti-inflammatory (Il 6 and Il 10), chemokines (Cxcl 12), antimicrobials and mucins (Def 1, Def 2, Lyz, and Muc 1) and antioxidant (Cat and Sod 2) genes were highly upregulated in Raw 264.7 cells exposed to SmPNPs. In brief, after 6 h of exposure to SmPNPs (0.5 and 1 mg/mL), there was a > 2-fold upregulation in the mRNA expression of Il 6 (2.36- and 2.88-fold), Il 10 (5.60- and 11.25-fold), Sod 2 (4.09- and 8.66-fold), Def 1 (3.13- and 8.28-fold), Def 2 (2.45- and 4.53-fold), and Muc 1 (2.01- and 4.95-fold), respectively. Similarly, there was a > 2-fold upregulation in the mRNA expression of Il 10 (3.43- and 6.18-fold), Sod 2 (3.16- and 4.82-fold), Def 2 (5.42- and 4.75-fold), Muc 1 (3.75- and 2.75-fold), and Cxcl l2 (2.74- and 2.29-fold) at 12 h after exposure to 0.5 and 1 mg/mL of SmPNPs, respectively. However, Lyz (3.63-fold) was significantly (*p* < 0.05) upregulated only in 1 mg/mL SmPNPs treated cells at 6 h, whereas Cat was upregulated but not significant in cells treated with 1 mg/mL of SmPNPs at both the time points (2.44-fold at 6 h and 2.56-fold at 12 h). Il 10, Sod 2, Muc 1, and Def 2, which were upregulated at 6 h, persisted (> 2.5-fold) even up to 12 h in cells exposed to 0.5 and 1 mg/mL of SmPNPs. Cxcl 12 (2.29-fold) showed significant upregulation only at 12 h after treatment with 1 mg/mL of SmPNPs. Furthermore, mRNA level of MyD88 was downregulated in cells treated with SmPNPs (0.5 and 1 mg/mL) at both the time points (0.38- and 0.36-fold at 6 h and 0.40- and 0.27-fold at 12 h, respectively). The mRNA levels of Tlr 4 were also downregulated in the cells treated with both concentrations at 6 h (0.24- and 0.36-fold), whereas values were reached to basal levels at 12 h (1.04- and 0.99-fold).

### 2.3. Reduction of Reactive Oxygen Species (ROS) Levels in Larvae upon Exposure to SmPNPs

We determined whether SmPNPs could reduce the levels of ROS induced upon *A. hydrophila* infection in larvae. The effect of two doses of SmPNPs in larvae infected with *A. hydrophila* was determined by quantifying the ROS levels using the 2′7′dichloro-dihydro-fluorescein diacetate (H_2_DCFDA) assay. Our data show that upon challenge with *A. hydrophila*, the relative fluorescence was higher in the control group larvae that were not treated with SmPNPs (head and pericardia, 266.3%; tail, 166.4%) compared to that in larvae exposed to SmPNPs (25 and 50 µg/mL) (Figure 2A). Furthermore, larvae exposed to 25 µg/mL of SmPNPs had the lower (*p* < 0.05) ROS levels (head and pericardia, 119.6%; tail, 69%) compared to that in the larvae exposed to 50 µg/mL of SmPNPs (head and pericardia, 149.6%; tail, 93.6%) (Figure 2B). Additionally, exposure of larvae to the lower dose (25 µg/mL) of SmPNPs was more effective in reducing ROS production induced by *A. hydrophila* infection than the exposure to the higher dose (50 µg/mL). Overall, the results revealed that SmPNPs have the ability to decrease oxidative stress, generated upon *A. hydrophila* infection.

### 2.4. Disease Resistance of Zebrafish Larvae and Adults Exposed to SmPNPs

We determined the disease resistance of zebrafish larvae exposed to SmPNPs (25 and 50 µg/mL) followed by immune challenge with *A. hydrophila*. The cumulative survival (53.3% and 33.3%) of larvae exposed to 50 and 25 µg/mL of SmPNPs was higher compared to that of larvae in the control (16.6%) group at 96 h post infection (hpi), respectively (Figure 3A). Furthermore, there was a dose-dependent increase in survival, and the cumulative survival % was significantly (*p* < 0.05) higher only in larvae exposed to 50 µg/mL of SmPNPs. The development of disease resistance in adult zebrafish fed a diet supplemented with SmPNPs (4%) was monitored after immune challenge with *A. hydrophila* at the end of the feeding trial at 6 weeks. The cumulative survival was 76.7% and 34.4% in SmPNPs supplemented and control groups at 96 hpf, respectively (Figure 3B). Moreover, the values were significantly (*p* < 0.05) different between the adults fed SmPNPs supplemented and control diets.

### 2.5. Transcriptional Profiling of Immune-Related Genes in Zebrafish upon SmPNPs Treatment

Transcriptional responses elicited by SmPNPs were analyzed under seven immune functional categories, namely, toll-like receptors, transcription factors, pro- and anti-inflammatory, chemokines, antimicrobials/enzymes and mucins, heat shock proteins, and antioxidant enzymes. Quantitative reverse transcription polymerase chain reaction (qRT-PCR) analysis was conducted for the above genes in SmPNPs exposed larvae at 5 days post fertilization (dpf) and in the gut of adult zebrafish that were fed SmPNPs supplemented diet for 6 weeks. Among the different functional categories nine gene transcripts were induced >1.5-fold in larvae exposed to the low dose (25 µg/mL) of SmPNPs (Figure 4). These gene transcripts were *muc5.2* (2.14-fold), *muc5.3* (2.08-fold), *hamp* (1.98-fold), *ctsd* (3.42-fold), *defβl1*(1.50-fold), *alp* (1.68-fold), *hsp90ab1* (1.77-fold), *cat* (2.57-fold), and *sod1* (1.64-fold). After exposure to higher dose of SmPNPs (50 µg/mL), 13 genes, namely *tlr2* (4.49-fold), *tlr4b* (4.59-fold), *tnfα* (2.53-fold), *cxcl18b* (1.57-fold), *ccl34a.4* (1.86-fold), *muc5.1* (2.79-fold), *muc5.2* (2.12-fold), *muc5.3* (4.50-fold), *hamp* (3.21-fold), *alp* (2.87-fold), *hsp90ab1* (3.68-fold), *cat* (1.80-fold)*,* and *sod1* (2.95-fold), were induced over 1.5-fold. Interestingly, among the eight genes induced at 25 µg/mL, the induction of seven (except *ctsd*) was maintained over 1.5-fold at 50 µg/mL. Moreover, *tlr2*, *tlr4b*, *tnfα*, *cxcl18b*, *ccl34a.4*, and *muc5.1* were only upregulated in larvae exposed to 50 µg/mL of SmPNPs.

Fifteen genes showed ≥ 1.5-fold upregulation in gut tissues of adult zebrafish fed SmPNPs; these genes were *tlr5b* (2.43-fold), *il1β* (3.46-fold), *tnfα* (1.53-fold), *cxcl8a* (3.47-fold), *cxcl18b* (4.08-fold), *ccl34a.4* (8.57-fold), *ccl34b.4* (3.52-fold), *lyz* (1.79-fold), *muc5.1* (5.78-fold), *muc5.3* (6.22-fold), *alp* (2.23-fold), *hsp90ab1* (1.67-fold), *hsp70* (3.27-fold), *cat* (2.61-fold)*,* and *sod1* (4.48-fold). Among the genes, which expression was induced ≥ 1.5-fold, nine (*tnfα*, *cxcl18b*, *ccl34a.4*, *muc5.1*, *muc5.3*, *alp*, *hsp90ab1, cat*, and *sod1*) were common in SmPNPs exposed larvae and SmPNPs (4%) fed adult zebrafish.

### 2.6. Effect of SmPNPs on the Expression of Alp and Hsp90 Proteins in Zebrafish Larave

To confirm the results of transcriptional responses of *alp* and *hsp90*, immunoblotting was performed with zebrafish larvae. Results showed that Alp and Hsp90 proteins were significantly (*p* < 0.05) upregulated in SmPNPs exposed (50 µg/mL) whole larvae compared to the control (Figure 5A). The relative intensities (normalized to β-actin) of Alp and Hsp90 were 2.00 and 1.90-folds for the samples treated with 50 µg/mL of SmPNPs, respectively (Figure 5B,C). Even though, the protein expression was slightly increased for Alp, none of the proteins were significantly (*p* > 0.05) expressed at 25 µg/mL. Therefore, the optimum concentration of SmPNPs for immunomodulation can be suggested as higher than 50 µg/mL.

### 2.7. Effect of SmPNPs on Dermal Wound Healing and Pigment Restoration in Adult Zebrafish

The effect of SmPNPs on dermal wound healing was assessed using an adult zebrafish wound model. Representative images taken during the wound healing process for the same individual fish of each group (vehicle and SmPNPs treated) are presented in Figure 6. A clear wound margin was identified initially at 2 days post wounding (dpw), and the average wound size was 3.30 ± 0.03 and 3.39 ± 0.05 mm^2^ for the vehicle and SmPNPs treated groups, respectively. To analyze the wound healing performance upon topical application of SmPNPs, wound healing percentage (WHP) and wound healing rate (WHR) at 7, 10, 14, and 24 dpw were calculated with respect to the respective values at 2 dpw. The microscopic images of wounds displayed a clear difference in the disappearance of the wound margins and re-establishment of pigments at the wound site over time. SmPNPs treated zebrafish showed rapid wound closure compared to the vehicle treated group at all the time points. WHP was significantly (*p* < 0.05) higher (48.90%) at 10 dpw compared to that in the vehicle treated group (38.80%) (Figure 6B). All zebrafish treated with SmPNPs showed completely healed wounds at 24 dpw. Similarly, the SmPNPs treated group had a higher wound healing rate at 7 and 10 dpw. (Figure 6C). Furthermore, we evaluated pigment restoration (melanin-based skin pigments/spots) during wound healing. Digital images representing the pattern of pigment restoration are shown in Figure 7A. SmPNPs treated fish had more restored pigment spots compared to the untreated fish during the whole wound healing process. Moreover, the estimated mean surface area pigment intensity was compared with a similar baseline (186.7 mm^2^) at each dpw, and only at 24 dpw, the values were significantly (*p* < 0.01) higher at the wound site; the value at 24 dpw in the SmPNPs treated group was 135.00 mm^2^ compared to 91.0 mm^2^ in the vehicle treated group (Figure 7B). Ultimately, almost complete restoration of melanin-based skin pigments on the black stripe pattern was observed in the SmPNPs treated group at 24 dpw.

### 2.8. Histological Assessment of the Effect of SmPNPs on Wound Healing

Histological analysis was performed after haematoxylin and eosin (H&E) staining to further confirm the effect of SmPNPs on wound healing. As shown in Figure 8, in unwounded muscle, the skin of zebrafish consisted of overlapping scales, which were wrapped by thin dermal fibroblasts and a multilayered epidermis. Because of the wound induced by laser, all epidermal cells, including scales, were damaged and removed, which can be clearly observed at 2 dpw (Figure 8). SmPNPs treated fish showed rapid re-epithelialization compared to the vehicle treated fish at 7 dpw. The well-forming granulation of tissues were first visible only in the SmPNPs treated group at 2 dpw, and showed increased infiltration of inflammatory cells whereas the same were observed in the vehicle treated group at 7 dpw. Moreover, the number of inflammatory cells was lower in the vehicle treated than in the SmPNPs treated wounded tissue at 2 dpw. At 7 dpw, the wound was completely re-epithelialized in the SmPNPs treated group with multiple cell layers of dense neoepidermis (red arrows). However, in the vehicle treated group, a thin layer of neoepithelium was observed with more number of inflammatory cells within the loose connective tissue on the wound surface. Quantitative analysis for neoepithelial thickness in healing wounds showed that SmPNPs treated group (53 µm) had higher thickness (*p* > 0.05) compared to vehicle treated group (18 µm). Moreover, semi- quantitative analysis at 2 and 7 dpw results further confirmed aforementioned rapid re-epithelialization, and well-forming granulation of tissues of SmPNPs treated group compared to the vehicle treated group (Appendix A).

### 2.9. Time-Course Transcriptional Analysis of Wound Healing in Adult Zebrafish upon SmPNPs Treatment

SmPNPs mediated wound healing was analyzed by quantifying the mRNA expression levels of genes under four different immune functional categories, namely, (i) pro-and anti-inflammatory genes (*tnfα*, *il1β*, and *il10*), (ii) chemokines (*cxcl18b*, *ccl34a.4*, and *ccl34b.4*); (iii) promotion of wound healing and tissue remodeling (*tgfβ1*, *mmp9*, *mmp13*, and *timp2b*), and iv) antioxidant enzymes (*sod1* and *cat*) using the muscle (as wounding site) and kidney (as immune functional organ) tissues. The relative fold-change in the mRNA expression levels of each gene was calculated in vehicle and SmPNPs treated groups based on the mRNA levels in the unwounded group. The time-course transcriptional profile of vehicle and SmPNPs treated groups compared to control is summarized in Figure 9.

#### 2.9.1. *tnfα*

The mRNA levels of *tnfα* in the muscle and kidney were highly induced as early as 1 dpw by 18.74- and 15.42-fold in the SmPNPs treated and by 6.50- and 4.00-fold in the vehicle treated group, respectively. Compared to the kidney, the muscle showed a relatively longer (1–14 dpw) upregulated profile of *tnfα* in the both vehicle and SmPNPs treated groups, whereas the kidney tissue had a down regulated profile in the SmPNPs treated group from 14 to 24 dpw.

#### 2.9.2. *il1β* and *il10*

The mRNA levels of *il1β* was increased in the muscle (5.75-fold) and kidney (3.66-fold) in the SmPNPs treated group compared to that in the unwounded control group at 7 and 1 dpw, respectively. In the kidney, both vehicle and SmPNPs treated groups showed downregulated or basal *il1β* mRNA levels, except at 1 dpw (3.66-fold) in the SmPNPs treated group. Moreover, > 2-fold mRNA expression level of *il10* was observed in the muscle in the SmPNPs treated group until 24 dpw. The *il10* was also upregulated in the kidney of SmPNPs and vehicle treated groups from 1 to 7 dpw, respectively.

#### 2.9.3. *cxcl18b*, *ccl34a.4* and *ccl34b.4*

mRNA levels of chemokines (*cxcl18b, ccl34a.4*, and *ccl34b.4*) were upregulated either at single or multiple time points in the SmPNPs treated group compared to that in the vehicle treated group. However, the mRNA level was different at each time point. Briefly, in the muscle, *cxcl18b* was significantly (*p* < 0.05) upregulated at 1 dpw in the SmPNPs treated group compared to its expression in the vehicle treated group. In contrast, significant (*p* < 0.05) increase in the expression of *cxcl18b* was observed at 2 and 7 dpw (137.60- and 3.24-fold, respectively) in the kidney tissue of SmPNPs treated group compared to that in the vehicle treated group (71.25- and 6.90-fold). The mRNA expression of *ccl34a.4* was significantly (*p* < 0.01) increased in the muscle of SmPNPs treated group by 2.38-, 4.54-, 4.08-, and 3.47-fold at 1, 2, 7, and 14 dpw, respectively. Moreover, *ccl34b.4* was increased (*p* < 0.05) by 2.02-fold at 1 dpw in the SmPNPs treated group. In contrast, in the kidney, *ccl34a.4* transcript level was increased significantly (*p* < 0.05) at 1 and 7 dpw (112.94 and 14.87-fold, respectively) and *ccl34b.4* at 2 dpw (4.25-fold) in SmPNPs treated group compared to the vehicle (1.67-fold) in kidney.

#### 2.9.4. *tgfβ1*

We compared the transcriptional levels of *tgfβ1* in the SmPNPs and vehicle treated groups with that in the unwounded control group. The mRNA level of *tgfβ1* was significantly higher (*p* < 0.05) in the muscle (7.77-, 12.71-, and 4.56-fold), and the kidney (7.19-, 4.44-, and 4.79-fold) at 1, 2, and 7 dpw, respectively. In the muscle, the level of *tgfβ1* reached a peak at 2 dpw in the SmPNPs treated (12.71-fold) and vehicle treated (4.32-fold) groups. The expression of *tgfβ1* decreased gradually with time. Interestingly, compared to that in the vehicle treated group, in the SmPNPs treated group, *tgfβ1* remained induced for a longer time in both the kidney and muscle tissues.

#### 2.9.5. *mmp9* and *mmp13*

Early induction of *mmp9* mRNA was observed in the muscle of the SmPNPs treated group compared to unwounded control and vehicle treated group. The mRNA level of *mmp9* was significantly (*p* < 0.05) induced in the muscle by 5.04-, 10.01-, and 2.32 -fold at 1, 2, and 7 dpw and in the kidney by 6.46-fold at 1 dpw. The mRNA levels of *mmp13* was increased in the muscle by 7.29-, 5.10-, and 4.62-fold at 1, 2, and 7 dpw and by 9.92-fold in the 24 dpw and in the kidney by 5.86-fold at 1 dpw compared to unwounded group.

#### 2.9.6. *timp2b*

To evaluate the possible role of tissue inhibitors of metalloproteinases (TIMP), we selected *timp2b* as a candidate gene in this study. The *timp2b* transcript was induced after wounding followed by SmPNPs treatment. In the muscle, mRNA level of *timp2b* was upregulated by 3.25- and 5.43-fold at 2 and 7 dpw compared to unwounded control group; however, similar elevated level was observed in the muscle of wounded and untreated group. Moreover, in the kidney, it was upregulated by 3.81- and 5.74-fold at 2 and 7 dpw, respectively, compared to unwounded control group. The mRNA level of *timp2b* was peaked at 7 dpw in both the muscle and kidney.

#### 2.9.7. *sod1* and *cat*

In the muscle, *sod1* transcript was significantly (*p* < 0.05) induced in both the vehicle and SmPNPs treated groups by 2.32- and 14.92-fold at 1 and 2 dpw, respectively, whereas it was induced 2.13-fold in the kidney at 1 dpw. Compared to the unwounded control, *cat* transcript was induced in the muscle of SmPNPs treated group up to 2.17-, 23.45-, and 2.25-fold at 1, 2, and 14 dpw, respectively, and up to 2.60-fold in the kidney at 1 dpw.

## 3. Discussion

Marine derived nanoscale biomaterials show completely new or improved biological properties owing to specific features, such as size, distribution, and morphology, when compared with larger particles, and can be changed to different forms with some modifications [20,21]. In order to reduce the particle size, mechanical method was notable to obtain the efficient biological activities via nano-derivatives, as whole effect is received to the nanoparticles from the biological materials itself [20]. In the present study, we successfully synthesized *S. maxima* derived SmPNPs by mechanical homogenization (sonication), and verified that the prepared SmPNPs were in the nano scale (125.3 nm), and smaller than SmP (202 nm). Besides, zeta potential of SmPNPs and SmP were −24.1 and −29.2 mv, respectively, which indicates the physical stability of SmPNPs. Both SmPNPs and, SmP showed the irregular shape according to the FE-SEM images; however, compared to SmP, SmPNPs were not aggregated, and, soluble in water, while SmP showed moderate solubility [11]. In general, the reduced particle size without any aggregation and, increased solubility of the materials may enhance the efficacy of uptake, delivery, and adsorption [21,22]. Therefore, we suggest that exploring the unknown biological properties of *S. maxima* pectin at the nanoscale (using SmPNPs) would be promising for further use of this microalga. Moreover, in this study, we have not used any chemical methods for the preparation of SmPNPs, and it could be assumed that the whole effect is solely inherent from *S. maxima* pectin. Next, we demonstrated the effect of SmPNPs on immunomodulation, stress tolerance, disease resistance against *A. hydrophila*, and wound healing in zebrafish, which is an emerging animal model for studies on nanomaterials because of optical transparency, low husbandry cost, and high degree of genomic homology to humans [23].

Toxicity assessment of nanomaterials involves the determination of potential hazards or bio-safety levels of test materials [24]. Therefore, prior to undertaking the aforementioned activities, we determined the toxicity of SmPNPs in vitro (using Raw 264.7 cells) and in vivo (using zebrafish larvae). The IC_50_ of SmPNPs for Raw 264.7 cells was determined to be 5.6 mg/mL. In vivo studies on zebrafish larvae revealed several malformations (pericardial oedema, axial malformation, head malformation, yolk sac oedma, and dead embryo) upon incubation with > 200 μg/mL of SmPNPs. However, SmPNPs exposed zebrafish embryos and larvae did not show significant (*p* > 0.05) mortality up to 200 µg/mL compared to the non-exposed control. Furthermore, SmPNPs did not affect the embryo hatchability below 200 µg/mL. As reported by us previously [13], the maximum non-toxic dose of SmP for zebrafish larvae is 50 µg/mL. Interestingly, in this study, SmPNPs exposed zebrafish embryos and larvae showed lower toxicity levels up to 200 µg/mL. Therefore, based on the LC_50_, we suggest that the safe concentration of SmPNPs for zebrafish would be ≤ 547 µg/mL. The low toxicity might be due to the particle size, particle shape, and surface functionalization of nanoparticles. [21]. Accordingly, 25 and 50 µg/mL were considered as non-toxic concentrations and were used as biologically safe doses for further studies.

Oxidative stress is an imbalance of the homoeostasis of redox reactions, and results in the increase in ROS levels [25]. ROS can oxidize the cellular components such as protein, lipids, DNA, etc., and ultimately cause the cell death via apoptosis [26]. In this study, the ROS levels were suppressed in larvae exposed to 25 and 50 µg/mL SmPNPs after *A. hydrophila* challenge. This suggests the role of SmPNPs in detoxifying the oxidative stress induced by bacteria. Although ROS production was reduced by SmPNPs, it was not in a concentration-dependent manner. This might have been due to the ROS quenching ability of SmPNPs. Fu et al. [27] explained that nano-toxicity is caused by the overproduction of ROS, which induces oxidative stress and results in the failure of maintenance of the normal physiological redox-regulated functions. Previous reports have shown that bio-polymers like SmP reduces oxidative stress induced by *Edwardsiella piscicida* [13], and that the extract of *Hizikia fusiforme* could act against hydrogen peroxide-induced oxidative stress in zebrafish [28]. Similarly, the results of our study imply that SmPNPs have the potential to reduce oxidative stress induced by *A. hydrophila*, which ultimately leads to high disease resistance.

*A. hydrophila* is a bacterial pathogen that causes motile septicaemia or haemorrhagic septicaemia, and results in high economic losses [29]. Commonly, immunostimulants are a group of compounds that trigger non-specific cellular defense mechanisms in animals. They are used as feed additives in the feed industry to enhance resistance to viral, bacterial, and parasite infections and improve fish health [30]. It has been reported that lipopolysaccharides (LPS) are effective in preventing the disease caused by *A. hydrophila* and stimulate the innate immune response in rainbow trout [31]. Similarly, Anderson et al. [32] showed that chitosan and N-acetylated chitin, could increase the protection against *A. salmonicida* infection by injecting or immersing brook trout (*Salvelinus fontinalis*). In this study, both SmPNPs exposed larvae and adult zebrafish showed disease resistance against *A. hydrophila* with increased survival and activation of innate immune genes. During the early stages of larvae, the innate immune system performs vital role until the adaptive immune system is developed. Therefore, the survival of larvae increased considerably with increased concentrations of SmPNPs, which was confirmed by the strong immune modulation via SmPNPs at the early stage of exposure. Previously, SmP was reported to exhibit an immune response against *A. hydrophila* challenge [13]. Interestingly, compared to SmP, SmPNPs showed a more effective activity, confirming that nanoparticles are more efficient than normal pectin (SmP). Overall, it could be suggested that SmPNPs not only enhance disease resistance in zebrafish against *A. hydrophila*, but also act as an important dietary supplement with immunomodulatory activity.

To further confirm the ROS detoxification and disease resistance properties of SmPNPs at the molecular level, transcriptional analysis of immune related genes was performed. Raw 264.7 cells exposed to safe doses of SmPNPs exhibited a concentration dependent alteration of the innate immune system by expressing the pro- and anti-inflammatory cytokine genes, Il 6 and Il 10, and genes of antioxidant enzymes, such as Sod and Cat, at 6 h after the exposure. Il-10 exerts a pleiotropic effect by regulating immune responses and inflammation through the production of pro-inflammatory cytokines [33]. IL-6 is the dominant mediator of the acute phase response, an innate immune mechanism that is triggered by infection and inflammation [34]. However, prolonged cytokine activation is not good for cellular homoeostasis; considering the increase in the levels of cytokines at 6 and 12 h, it could be suggested that cellular homeostasis is maintained by SmPNPs without showing detrimental effects.

Disease resistance of zebrafish larvae against the *A. hydrophila* can also be further explained by the upregulation of different innate immune-related genes (*tlr2*, *tlr4b*, *tnfα*, *muc5.1*, *muc5.2*, *muc5.3*, *hamp*, *ctsd*, and *sod1)*. Interestingly, most of these genes were upregulated at a high dose (50 µg/mL) of SmPNPs, which was also confirmed by in vitro SmPNPs treatment. Furthermore, *S. platensis* derived C-phycocyanin showed immunomodulation by increasing the secretion of inflammatory cytokines, TNF-α, IL-1β, and IL-6, in murine macrophages [35]. Chemokines act as chemotactic cytokines and activate and regulate the migration and location of immune cells in infected or damaged organs or tissues [36]. In fact, we observed the induction of all the tested chemokine genes, such as *cxcl18a*, *cxcl18b*, *ccl34 a.4*, and *ccl34 b.4* in adult fish exposed to SmPNPs. The mRNA expression of *muc5.3*, *alp*, *lyz*, and *hsp70* was considerably induced in the gut of adult zebrafish that were fed SmPNPs supplemented (4%) diet. Similarly, antioxidants like *cat* and *sod1* showed higher increase in both larvae that were exposed and adults that were fed SmPNPs compared to the control, which strongly confirmed the in vitro results and the critical role of SmPNPs in the zebrafish innate immune system.

Heat shock protein transcripts upregulate under high temperature stress, and play an important role in heat stress response in living cells. [37]. The members of the heat shock protein 90 (Hsp90) family mediate stress signal transduction and play a vital role in the controlling the normal growth of cells in all living organisms [38]. Intestinal alkaline phosphatase (IAP) is important for modulating bacterial LPS induced inflammation by detoxifying LPS [39]. Furthermore, IAP showed anti-inflammatory effects in a Toll-like Receptor-4 (TLR-4) dependent manner [40]. Our results for protein expression showed elevated expression of Hsp90 and Alp in the 50 µg/mL of SmPNPs group, which further validated the upregulated *hsp90* and *alp* mRNA expression upon SmPNPs treatment. In agreement with our results, Chandrarathna et al. [20] also reported elevated expression of IAP at the mRNA and protein levels, implying the anti-inflammatory action of MSmP and MSmPNPs. Collectively, these results further confirmed the stress tolerance and anti-inflammatory properties of SmPNPs.

Wounding is the result of disruption of the normal anatomic structure and function of a tissue [41]. After injury, the integrity of skin and other tissues must be promptly restored to maintain its functions and resident skin cells, peripheral blood mononuclear cells, extracellular matrix, growth factors, and regulatory molecules participate in the wound healing process [42]. Recent studies on wound healing have reported that alkaloids, essential oils, flavonoids, tannins, terpenoids, saponins, and phenolic compounds derived from natural products can be used as wound healing agents [43]. Moreover, pectin has been previously reported to possess bioactivities, such as promotion of the proliferation of B cells and the secretion of interleukin-1β by macrophages during wound healing. Smith et al. and Munarin et al. [44,45] described the physicochemical properties of pectin, which provide several benefits as wound dressing materials owing to their hydrophilicity, removal of exudates, and maintenance of an acidic pH, which is expected to act as a barrier against bacteria or fungi. In our study, SmPNPs treated adult zebrafish showed rapid wound closure compared to that in the vehicle treated group at each time point. Furthermore, compared to the vehicle treated group, WHR was increased in the SmPNPs treated group at 7 and 10 dpw. The relationship between SmPNPs treatment and pigment restoration during wound healing also confirmed that SmPNPs treated fish had higher number of restored pigment spots compared to the untreated zebrafish during the whole wound healing process until 24 dpw, which showed significant (*p* < 0.01) pigment restoration at 24 dpw compared to that in the vehicle treated zebrafish. Histological analysis also supported the above results and confirmed that SmPNPs are wound healing agents, which act through the development of the neoepidermis, increased inflammatory cell density, and low inflammation. Seo et al. [46] reported that topical application and immersion in AgNPs had a positive effect on wound healing. Interestingly, this study demonstrated that topical application of SmPNPs promotes wound closure and pigment restoration and induces the initial protective mechanism of wound opening without incorporation of metal or any other modifications during compound preparation. However, the results obtained in this study, and our previous report of SmP [14], have no remarkable differences in wound healing activity.

However, to understand the mechanism underlying the effects of SmPNPs on wound healing at the molecular level, transcriptional response of genes involved in the wound healing process was analyzed. Tumour necrosis factor-alpha (TNF-α) is an inflammatory cytokine produced by macrophages in the inflammation phase during the wound healing process. It has been shown to exert a wide variety of biological effects, leading to necrosis and apoptosis, and responds to a diverse range of signaling events within cells [47]. TGF-β is a multifunctional growth factor that exerts pleiotropic effects on wound healing by regulating cell proliferation, migration, differentiation, and immune modulation [48]. The release of TGF-β1 during the response to injury plays a critical role in the chemotaxis of macrophages and fibroblasts to the wound site. TGF-β released by T cells has been shown to induce collagen synthesis and angiogenesis [49]. IL-10 is a cytokine produced by a variety of cell types, including T cells, monocytes, and macrophages, and is capable of producing IL-10 after injury [50]. Moreover, previous studies have shown that matrix metalloproteinases (MMPs) play a significant role in wound healing. MMP13 plays a role in angiogenesis, keratinocyte migration, and contraction in wound healing. MMP-9 is also involved in keratinocyte migration [51]. In this study, several genes of pro and anti-inflammatory cytokines *(tnfα, il1β,* and *il10*), chemokines (*ccl34a.4*)*,* growth factor (*tgfβ1*), MMPs (*mmp9* and *mmp13*), *timp2b, sod1*, and *cat* were highly upregulated in SmPNPs treated groups at 1 and 2 dpw in the muscle and kidney at both or either of the time points. Interestingly, apart from early responses, *mmp13* was upregulated (9.92-fold) at 24 dpw during the late stage of wound healing, suggesting that SmPNPs enhance the expression of *mmp13* and lead to rapid and efficient wound healing. Overall, these findings confirm that SmPNPs may engage in the regulation of the entire process and immune modulation during wound healing.

In conclusion, our study provides novel insights into the positive effects of SmPNPs on immunomodulation, stress tolerance, disease resistance, and dermal wound healing. Moreover, the results obtained in this study concur with those of our previous studies on SmP and MSmP, which modulate the immune and gut microbiota in zebrafish and mice, respectively. Hence, the present study warrants further investigations to understand the underlying mechanisms and multiple functions.

## 4. Materials and Methods

### 4.1. Preparation and Characterization of SmPNPs

*S. maxima* derived pectin (SmP) was provided by Korea Institute of Ocean Science and Technology (KIOST), Jeju Special Self-Governing Province, Korea [8]. For the preparation of SmPNPs, SmP (100 mg) was dissolved in autoclaved deionized water (10 mL), and sonicated under amplitude 30%, 10:10 sec pulse at 4 °C for 30 min. Then the sonicated solution was centrifuged (Combi 514R, Hanil Science, Gimpo, Korea) at 3500 rpm for 15 min and supernatant (SmPNPs) was used for physiochemical characterization. Particle size distribution and zeta potential of SmPNPs were determined by Zetasizer S-90 Malvern instrument (Malvern, Worcestershire, UK). The morphology of SmPNPs was determined as previously described by the field emission scanning electron microscopy (FE-SEM, S-4800, Hitachi, Tokyo, Japan) [52] and field emission transmission electron microscopy (FE-TEM, Model Tecnai G2 F30 S-Twin, FEI, Atlanta, GA, USA) operating at 300 keV [53].

### 4.2. Cell Culture and In Vitro Cytotoxicity of SmPNPs

Raw 264.7 cell line was obtained from Dr. Jong-Soo Lee, College of Veterinary Medicine, Chungnam National University, Korea. Raw 264.7 cells were cultured in Dulbecco’s modified Eagle’s medium supplemented with 10% (*v/v*) fetal bovine serum (WELGENE Inc., Daegu, Korea) and antibiotic–antimycotic solution (Gibco™, GrandIsland, NY, USA) at 37 °C in a 5% CO_2_ incubator. Cell cytotoxicity was determined by MTT reduction assay. In brief, cells were seeded at a density of 2 × 10^5^ cells per well in a 96-well plate and were incubated at 37 °C with different concentrations (1–8 mg/mL) of SmPNPs for 24 h. For the control, autoclaved distilled water was used. After incubation, the medium was replaced, and 10 µL of MTT solution (5 mg/mL) was added to each well and incubated for 4 h at 37 °C. The resulting formazan crystals were dissolved in 50 µL of dimethyl sulfoxide (DMSO; Sigma Aldrich, Saint Louis City, MO, USA). The absorbance was measured at 595 nm using a microplate spectrophotometer (Bio Rad Laboratories, Inc., Richmond, CA, USA).

### 4.3. Zebrafish Husbandry and Assessment of In Vivo Toxicity in Zebrafish Embryos

All experiments with zebrafish were conducted in accordance with the approved guidelines and regulations of the Animal Ethics Committee of Chungnam National University, Korea. Wild type (AB) zebrafish were provided by the Zebrafish Center for Disease Modeling (ZCDM), Korea. The adult zebrafish (4–8 months) were maintained in an automated water circulation system with a 14 h light: 10 h dark photoperiod at 28 ± 5 °C. After successive mating of fish, embryos were collected at 1 hpf and raised in embryonic medium (EM) containing 60 mg/L aquarium salt. In vivo toxicity of SmPNPs was evaluated using healthy embryos (1 hpf) in 6-well plates (10 embryos per replicate) by exposure to different concentrations of SmPNPs (0–600 µg/mL), which were diluted in EM, until 96 hpf at 28 °C. Experiments were carried out in triplicate and EM was used as a control. The cumulative mortality percentage (%) of SmPNPs exposed larvae was calculated and compared with that for the non-exposed control group. Two concentrations (25 and 50 µg/mL) of SmPNPs were selected as non-toxic to determine the effects of SmPNPs in further assays.

### 4.4. Determination of ROS Levels Induced by A. hydrophila Infection after Exposure to SmPNPs

Reduction of *A. hydrophila* induced ROS levels upon exposure to SmPNPs was quantified using the fluorescent dye, H_2_DCFDA (Sigma-Aldrich, St Louis, MO, USA), as described by Lakey et al. [54] and Edirisinghe et al. [13]. Briefly, the experiment was conducted in four groups (*n* = 10): (1) negative control (without SmPNPs exposure or *A. hydrophila* challenge), (2) control (without SmPNPs exposure, but with *A. hydrophila* challenge), (3) 25 µg/mL of SmPNPs exposed and *A. hydrophila* challenged, and (4) 50 µg/mL of SmPNPs exposed and *A. hydrophila* challenged. The levels of ROS in larvae in all the groups were quantified at 48 hpi by exposure to H_2_DCFDA (50 µM) for 30 min in the dark at 28 °C. Excess dye was removed by washing with phosphate-buffered saline (PBS) three times. ROS generation in whole larvae (head with pericardia and tail) was imaged using a microscope (Nikon SMZ1000, Tokyo, Japan) connected to a Moticam Pro 205A microscope camera (Motic Deutschland GmbH, Wetzlar, Germany), equipped with a fluorescence adaptor (NIGHTSEA, Lexington, MA, USA) (Wave length of excitation: 440–460 nm). Quantitation of ROS was done using the ImageJ software (ImageJ, version 1.6, Bethesda, MD, USA) and percentage intensity of fluorescence was calculated relative to the untreated control.

### 4.5. SmPNPs Supplemented Diet Feeding Trial with Zebrafish

To investigate the disease resistance and immunomodulatory effects of SmPNPs on adult zebrafish, a SmPNPs (4%) supplemented diet was prepared by mixing SmPNPs with a commercial feed (on a weight basis). The prepared SmPNPs (as described in Section 4.1) were thoroughly mixed with the commercial feed (on a dry basis), and the pelleted diet was oven-dried overnight. Briefly, zebrafish (*n* = 140, mean weight 0.25 ± 0.03 mg) were placed in four 20 L tanks (*n* = 35) to maintain two replicates per group. Prior to the feeding trial, fish were acclimatized for 1 week. The feeding trial consisted of two groups: (1) control (100% commercial fish feed) and (2) SmPNPs (4%) supplemented feed fed. The formulation was orally administered to adult fish for 6 weeks. After feeding was terminated, six fish from each group (three fish/replicate) were randomly taken, and their gut tissue was aseptically isolated. The collected gut tissues were snap frozen in liquid nitrogen and, stored at −80 °C until RNA isolation for immune gene expression analysis. The remaining fish (at 6 weeks) were used for immune challenge with *A. hydrophila*.

### 4.6. Immune Challenge of Larvae and Adult Zebrafish with A. hydrophila

Disease resistance and immunomodulation capacities of SmPNPs were investigated by immune challenge of SmPNPs exposed larvae for 5 days and adult zebrafish fed SmPNPs supplemented diet for 6 weeks followed by *A. hydrophila* challenge. In brief, the experiments on larvae were conducted in 6 well plates and larvae (at the 2 hpf stage) were exposed to 25 and 50 µg/mL SmPNPs until 5 dpf. Three replicates (*n* = 10) were used for each treatment, and the control group was maintained under the same conditions without exposure to SmPNPs. Thermal stress (34 °C for 2 h) was given to all the larvae prior to the immune challenge to facilitate the induction of the stress condition. After the thermal stress, larvae were exposed to *A. hydrophila* (3.3 × 10^7^ CFU/mL) and culture plates were maintained at 28 °C. Adults fed the SmPNPs supplemented diet were challenged by intraperitoneal (i.p.) injection (20 µL) of *A. hydrophila* (1.56 × 10^6^ cells/fish) [13]. Mortality was recorded for both larvae and adults every 12 h, and cumulative survival percentage was calculated accordingly.

### 4.7. Transcriptional Analysis of Immunomodulatory Genes upon SmPNPs Treatment

Transcriptional analysis was performed for SmPNPs exposed Raw 264.7 cells, and zebrafish larvae and adults fed for SmPNPs supplemented diet. To collect samples, experiments were conducted as follows: Briefly, Raw 264.7 cells (2.0 × 10^5^ cells/mL) were seeded in a 6-well plate and allowed to adhere overnight (three replicates/treatment). Cells were maintained as described in Section 4.2. After replacing the medium, the cells were exposed to 0.5 and 1 mg/mL of SmPNPs at 37 °C for 24 h in a 5% CO_2_ incubator. Following aspiration of the medium, the cells were washed with PBS and centrifuged at 1000 rpm for 2 min. After complete removal of the supernatant, the cell pellet was stored at −80 °C, until RNA isolation. To investigate the immunomodulation effect, SmPNPs exposure trials with larvae were conducted in 6-well plates in three groups: (1) control and (2) 25 and (3) 50 µg/mL of SmPNPs exposure groups; the experiments were conducted in triplicates (with 15 embryos/replicate in 10 mL of embryonic medium). For the respective treatment groups, sterilized SmPNPs (1 mg/mL working solution) were added at final concentrations of 25 and 50 µg/mL. To the control group, 10 mL of EM was added, and both the treatment and control groups were incubated at 28 °C. At 5 dpf, 40 larvae from each group exposed to SmPNPs were randomly collected in microcentrifuge tubes (2 mL). The collected larvae were snap frozen in liquid nitrogen and stored at −80 °C until RNA isolation. Total RNA was isolated from Raw 264.7 cells and larvae exposed to SmPNPs, and from the gut tissue of adult zebrafish fed a diet supplemented with SmPNPs (4%) using TRIzol reagent (Sigma-Aldrich, St Louis, MO, USA) according to the manufacturer’s instructions. The concentrations of all the extracted RNA samples were measured using a NanoDrop One (Thermo Scientific, Waltham, MA, USA). Total RNA (2.5 μg) was used to synthesize the first-strand cDNA using PrimeScript™ first-strand cDNA synthesis kit (TaKaRa, Tokyo, Japan) following the manufacturer’s instructions. The synthesized cDNA samples were further diluted 40x and stored at −20 °C for qRT-PCR analysis. qRT-PCR was performed to analyze the gene expression in SmPNPs treated Raw 264.7 cells, larvae, and adults with their respective controls using the Thermal Cycler Dice Real Time System (TaKaRa, Tokyo, Japan). The mRNA expression was normalized to mouse Gapdh and zebrafish *β-actin*, and relative expression was determined using the 2^−ΔΔCt^ method [55]. Fold changes were calculated by dividing the average relative expression (fold) in the SmPNPs treatment with that in the respective control. The description of the gene-specific primer sequences and conditions is presented in Appendix A.

### 4.8. Immunoblot Analysis of Heat Shock Protein (Hsp90) and Alkaline Phosphatase (Alp)

Immunoblot analysis of SmPNPs treated (25 and 50 µg/mL) zebrafish larvae (5 dpf) (Section 4.7) was performed as described previously [20,56]. Briefly, the samples were homogenized (1 min) with 300 μL of ice-cold lysis buffer, (pH 7.6) (ProEX^TM^ CETi, TransLab Inc., Daejeon, Korea). The respective homogenates were centrifuged at 12,000 rpm for 10 min at 4 °C. Bradford protein assay (Bio-Rad Laboratories, Inc., Hercules, CA, USA) was performed to quantify the protein levels in each sample. Samples were then denatured with 2X Laemmli sample buffer (Sigma Aldrich, St Louis, MO, USA), at 100 °C, for 5 min, and an equal amount of protein (35 µg) was loaded onto 10% Sodium Dodecyl Sulfate – Poly Acrylamide Gel Electrophoresis (SDS-PAGE) and electrophoresed at 80 V for 30 min, and subsequently at 110 V for 1 h. The proteins were transblotted onto Polyvinylidene Difluoride (PVDF) membranes, which were then blocked with 5% bovine serum albumin (BSA). The membranes were incubated with Hsp90, Alp, and β-actin primary antibodies (1:3000) diluted in 5% BSA overnight at 4 °C. Thereafter, the membranes were washed three times with Tris-buffered saline containing Tween 20 (TBST) and incubated for another 1 h with respective secondary antibodies (1:3000) diluted in 5% BSA for 1 h at room temperature (25 °C). Detection of specific proteins was carried out using a chemiluminescence detection system (Fusion Solo S, Vilber, Lourmat, France). The band intensities were quantified using the Evolution-CAPT software (FUSION software user and service manual-v17.03, Vilber Company, San Sebastiàn, Spain), and normalized with respect to the expression of β-actin to obtain the relative protein expression. Three independent experiments were carried out to quantify the average expression of Hsp90 and Alp.

### 4.9. Wounding and Topical Treatment of Adult Zebrafish with SmPNPs

Four-month-old, (average body weight 0.45 + 0.05 g), uniform-sized, 162 zebrafish were divided (27 fish/replicate) into three groups: (1) control; (2) vehicle (wounded, treated with sterile deionized water); and (3) SmPNPs (wounded, treated with SmPNPs). Fish were acclimatized for 1 week prior to the experiment. They were anaesthetized using 0.12% (*w/v*) tricaine, and wounds were created with a laser beam (150 mA for 5 s) on a dark stripe at the left flank anterior to the anal and dorsal fins. To determine the effect of SmPNPs on wound healing, SmPNPs (600 µg/fish) were directly applied to the wound site in the SmPNPs group at 0, 1, 2, 4, 6, and 10 dpw. The vehicle group was treated with sterile deionized water, and all the fish were kept outside for 4 min after the treatments. The group with no wounds (Control) was also anaesthetized and kept outside for the same duration to simulate the conditions created for the vehicle and SmPNPs treated groups.

### 4.10. Effect of SmPNPs on Wound Closure and Pigment Restoration

To analyze the wound healing performance upon SmPNPs (topical application) treatment, we examined the visual wound closure, by measuring the wound size using microscopic images and analyzed the intensity of the melanin based pigment development [14,57] from 2 to 24 dpw. WHP and WHR were quantified as described previously [14]. In brief, wounded fish (*n* = 8) were selected at 2 dpw from each group and kept in individual tanks (500 mL). The wound site of each individual fish was imaged using a digital camera (Leica^®^ KL300 LED, Leica microsystems, Wetzlar, Germany, Wetzlar, Germany) connected to a stereo-microscope (Leica^®^ S8 APO, Leica microsystems, Wetzlar, Germany) at 2, 7, 10, 14, and 24 dpw. When the original wounded area was no longer distinguishable or was completely pigmented, the wound was completely healed. The pigment restoration was analyzed using the ImageJ software (ImageJ, ver. 1.6, Bethesda, MD, USA). In brief, individuals from each group were measured to find the difference between the normal and depigmented pale skin color of wounded fish, and then the pigment development on the dark stripe of the wound site was analyzed by normalizing to a similar baseline at 7, 10, 14, and 24 hpw.

### 4.11. Histological Analysis of the Effect of SmPNPs on Wound Healing

Tissue regeneration upon SmPNPs treatment was examined by histological analysis [14,57] at 2 and 7 dpw. Briefly, two fish from each replicate (control, vehicle, and SmPNPs treated) at 2 and 7 dpw were anaesthetized with an overdose of tricaine, and the wounded muscle tissues were surgically removed. Each tissue was soaked in PBS and fixed in 10% neutral buffered formalin. For sectioning, muscle samples were decalcified in 0.5 M EDTA (in neutral pH base) for 3 days at 28 °C. After the 12 h washing step, tissues were dehydrated by passing through an ascending graded series of alcohol and cleared with xylene in a Semi-enclosed Benchtop Tissue Processor (Leica^®^ TP1020, Leica microsystems, Nussloch, Germany, Germany). Then the tissues were embedded in paraffin wax (Leica^®^ EG1150 Tissue Embedding Center, Nussloch, Germany) and sectioning was performed (Leica^®^ RM2125 microtome, Leica microsystems, Nussloch, Germany). Four micrometer thick sections were stained with H&E (Sigma Aldrich, St Louis, MO, USA) according to the standard protocol. Stained tissues of each group were imaged using a digital camera (LEICA^®^ DCF450-C, Leica microsystems, Wetzlar, Germany) connected to a microscope (LEICA^®^ DM 3000 LED, Leica microsystems, Wetzlar, Germany). Finally, images were quantitatively analyzed for major histological parameters of wound healing. Neoepithelial thickness of healing wounds (control, vehicle and SmPNPs treated groups) were calculated as average along the wounded site. Histological images at 2 and 7 dpw were semi-quantitatively analyzed for histological parameters (abundance of inflammatory cells, scab, necrosis and slough, re-epithelialization and formation of granulation tissue) of wound healing by an observer who was blinded to the prior treatment of the fish.

### 4.12. Transcriptional Analysis of Wound Healing in Adult Zebrafish upon SmPNPs Treatment

In order to analyze the mRNA expression levels of different wound healing related immune genes, muscle and kidney tissues were collected from three zebrafish for each replicate (control, vehicle and SmPNPs treated) at 1, 2, 7, 14, and 24 dpw. RNA isolation, cDNA synthesis and qRT-PCR analysis were performed as described in Section 4.7. Gene specific primers and housekeeping genes were listed in Appendix A.

### 4.13. Statistical Analysis

Data were analyzed by one-way and/or two-way analysis of variance (ANOVA) to find out the overall significance between the experimental groups and/or time points. Bonferroni post hoc and/or unpaired two tailed *t*-test was conducted to compare the average means of the controls and the treatments. The cutoff for significance level was set as *p* < 0.05. The data were analyzed using GraphPad Prism software version 5 (GraphPad Software Inc., La Jolla, CA, USA).

## Figures and Tables

**Figure 1 marinedrugs-18-00556-f001:**
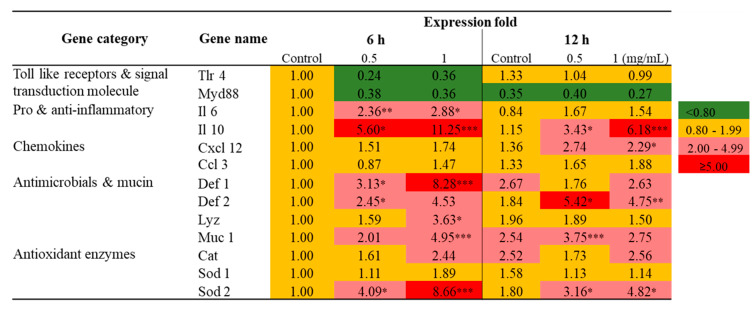
Transcriptional profiling of immune related genes in Raw 264.7 cells exposed to SmPNPs (0.5 and 1 mg/mL) for 6 and 12 h. Data are expressed as means plus or minus standard deviation (± SD) of triplicate samples. Asterisk (*) marks indicate statistical significance compared to SmPNPs treated vs. untreated control (One-way ANOVA * *p* < 0.05, ** *p* < 0.01, and *** *p* < 0.001). Basal level, upregulated, and down regulated expressions are considered as 0.80–1.99-fold, ≥ 2.0-fold, and < 0.80-fold, respectively.

**Figure 2 marinedrugs-18-00556-f002:**
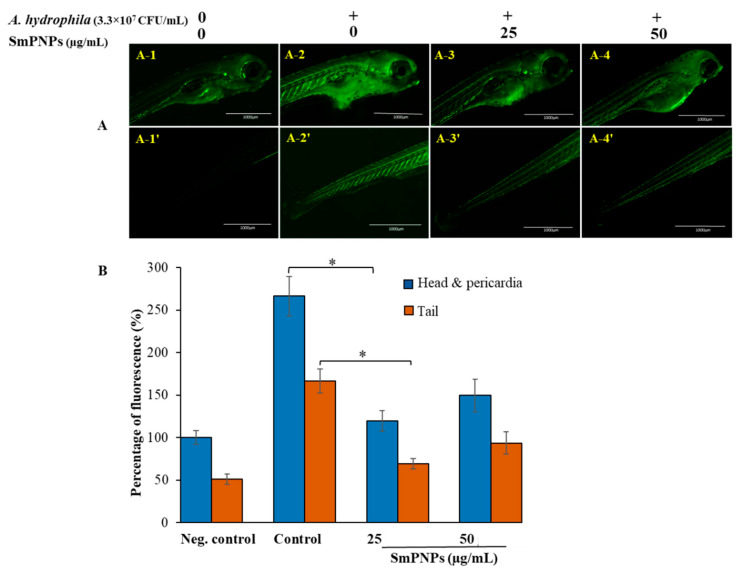
ROS detoxification effect of SmPNPs on *A. hydrophila* challenged zebrafish larvae. (**A**) Fluorescence images of head, pericardia, and tail areas were detected as follows: A-1, A-1′; Negative control (without SmPNPs exposure or *A. hydrophila* challenge), A-2, A-2′; Control (without SmPNPs exposure, but with *A. hydrophila* challenged), A-3, A-3′; SmPNPs (25 µg/mL) exposed and *A. hydrophila* challenged, A-4, A-4′ SmPNPs (50 µg/mL) exposed and *A. hydrophila* challenged. (**B**) The graph represents the percentage of fluorescence intensity compared to the untreated control group. Values were presented as means plus or minus standard error (± SE), and the asterisk (*) marks are used to indicate the significant difference compared to the respective controls (One-way ANOVA, unpaired two-tailed *t*-test, * *p* < 0.05).

**Figure 3 marinedrugs-18-00556-f003:**
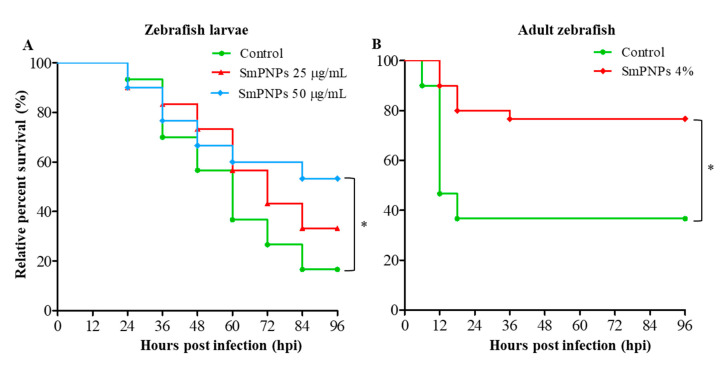
Disease resistance of SmPNPs exposed larvae and adult zebrafish against *A. hydrophila* infection. Immune challenge of (**A**) SmPNPs (25 and 50 µg/mL) exposed larvae infected with *A. hydrophila* (3.3 × 10^7^ CFU/mL) and (**B**) SmPNPs supplemented (4%) diet fed adults that intraperitoneal injected with *A. hydrophila* (1.5 × 10^6^ cells/fish). Significant differences (Kaplan-Meier, Wilcoxon; * *p* < 0.05) between the SmPNPs exposed vs. control were marked with asterisk (*) marks in graphs.

**Figure 4 marinedrugs-18-00556-f004:**
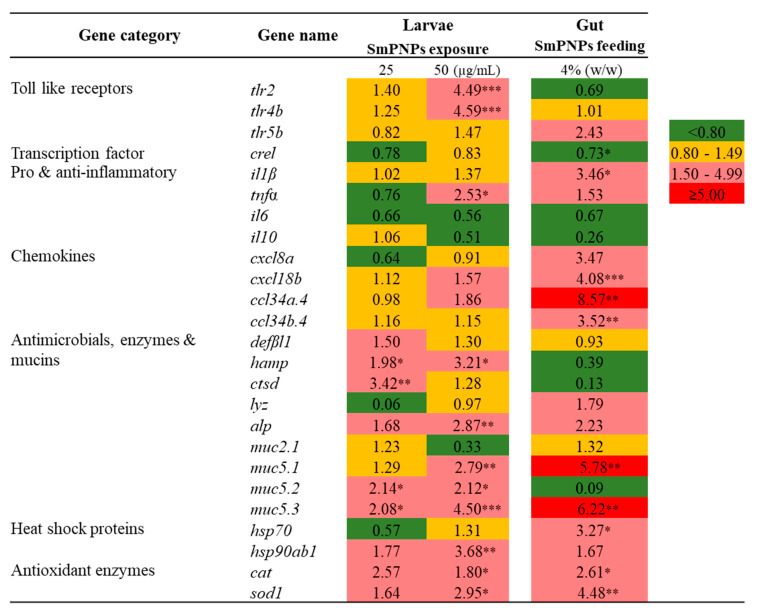
Transcriptional profiling of immune-related genes in zebrafish larvae exposed to SmPNPs (25 and 50 μg/mL) for 5 days and SmPNPs supplemented (4%) diet fed adults for 6 weeks. Data are expressed as means plus or minus standard deviation (± SD) of triplicate samples. Asterisk (*) marks indicate statistical significance compared to SmPNPs treated vs. non-treated control (One-way ANOVA * *p* < 0.05, ** *p* < 0.01, *** *p* < 0.001). Basal level, upregulated and down regulated expressions are considered as 0.80–1.49, ≥ 1.50-fold, and < 0.80-fold, respectively.

**Figure 5 marinedrugs-18-00556-f005:**
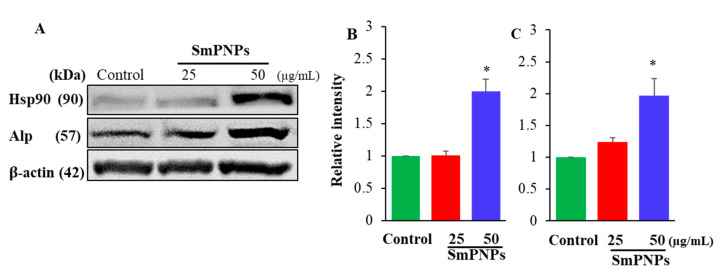
Immunoblotting of heat shock protein 90 (Hsp90) and alkaline phosphatase (Alp) expressions of SmPNPs exposed zebrafish larvae. (**A**) Immunoblots for Hsp90, Alp, and β-actin (loading control) were performed using whole zebrafish larvae (5 dpf) lysates of SmPNPs exposed (25 and 50 μg/mL) and control larvae. Quantitative analysis of (**B**) Hsp90 and (**C**) Alp expressions. Expression folds were normalized to β-actin. Values were presented as means plus or minus standard error (± SE), and the asterisk (*) marks are used to indicate the significant difference compared to the respective controls (One-way ANOVA, unpaired two-tailed *t*-test, * *p* < 0.05).

**Figure 6 marinedrugs-18-00556-f006:**
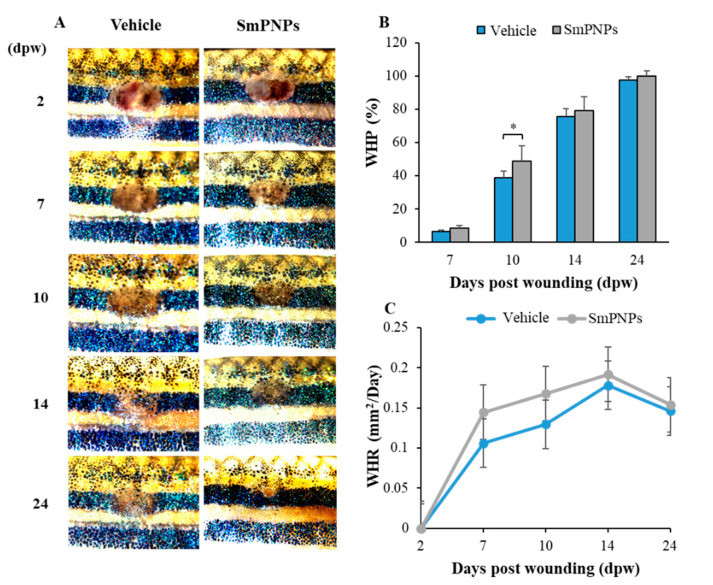
Wound healing effects of SmPNPs on adult zebrafish. (**A**) Representative images of wound healing process. (**B**) Wound healing percentage; WHP and (**C**) Wound healing rate; WHR at 7, 10, 14, and 24 dpw. On each day, WHP was calculated based on the wound size at 2 dpw. Error bars represent the means plus or minus standard deviation (± SD); (Unpaired two-tailed *t*-test, ** p* < 0.05, *n* = 8).

**Figure 7 marinedrugs-18-00556-f007:**
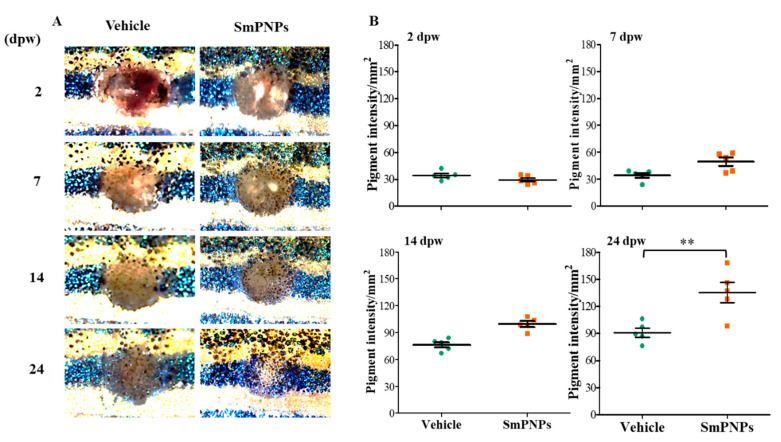
Effect of SmPNPs on pigment restoration during the wound healing. (**A**) Respective images showing the pigment development and accumulation on the wound site, and (**B**) Quantitative analysis of pigment development at 2, 7, 14, and 24 dpw. Error bars represent the means plus or minus (± SD) standard deviation (Unpaired two-tailed *t*-test, ** *p* < 0.01, *n* = 5).

**Figure 8 marinedrugs-18-00556-f008:**
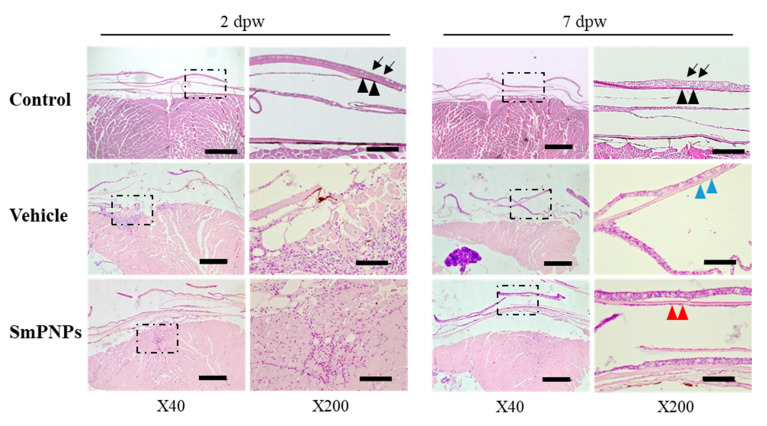
Effect of SmPNPs on wound healing by histology assessment; Hematoxylin and Eosin (H&E) staining of transverse sections of control (unwounded), vehicle (wounded; vehicle treated), SmPNPs (wounded; SmPNPs treated) muscle tissues at 2 and 7 dpw. Normal epidermis (black arrows), dermis and scale (arrowheads), persistent inflammation and thin layer of neoepithelium (blue arrows), completely re-epithelialized with a neoepidermis of multiple cell layer (red arrows) are shown in respective figures. Scale bar × 40 = 500 µm, × 200 = 100 µm.

**Figure 9 marinedrugs-18-00556-f009:**
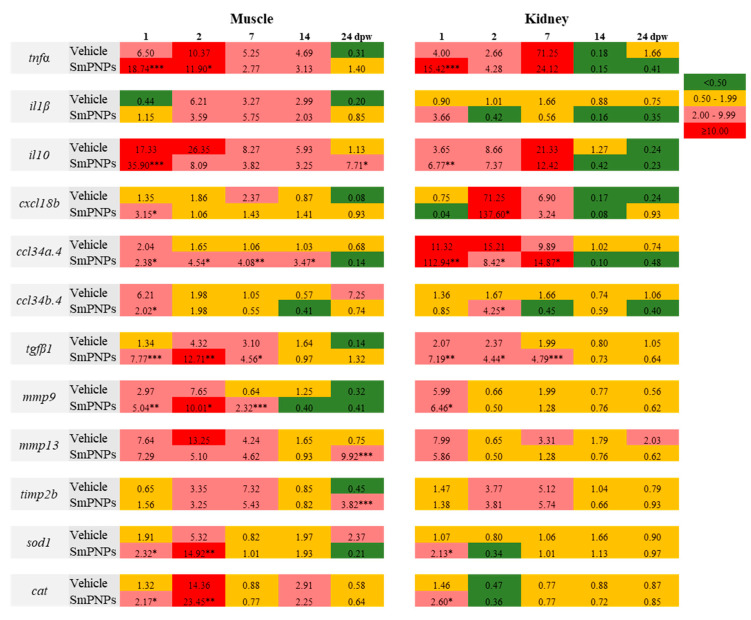
Time-course transcriptional analysis of genes related to wound healing in muscle and kidney of zebrafish. The wounded fish had been treated with vehicle and SmPNPs at different time points (1, 2, 7, 14, and 24 dpw) were analyzed. Data are expressed as means plus or minus standard deviation (± SD) of triplicate samples; (Unpaired two-tailed *t*-test * *p* < 0.05, ** *p* < 0.01, *** *p* < 0.001). Relative fold-change in gene expression was determined by dividing the average relative expression of each individual at selected time points (*n* = 3) by the average relative expression of control group at day 1 (*n* = 3). Basal level, upregulated and downregulated expressions are considered as 0.50–1.99, ≥ 2.0-fold and < 0.5-fold, respectively.

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
