# Peer review of "Spirulina maxima Derived Pectin Nanoparticles Enhance the Immunomodulation, Stress Tolerance, and Wound Healing in Zebrafish"

_marinedrugs, 2020, doi:10.3390/md18110556_

Round 1

Reviewer 1 Report

The manuscript entitled “Spirulina maxima Derived Pectin Nanoparticles Enhance the Immunomodulation, Stress Tolerance and Wound Healing in Zebrafishnds” presents the evaluation of immunomodulatory, stress tolerance, disease resistant and wound-healing activities of Spirulina maxima derived pectin nanoparticles. 

Major revisions should be made in order to be published in Marine Drugs journal, and the manuscript should be completed and/or modified taking into account the suggestions from the attached file.

Author Response

Reviewer #1: The manuscript entitled “Spirulina maxima Derived Pectin Nanoparticles Enhance the Immunomodulation, Stress Tolerance and Wound Healing in Zebrafish” presents the evaluation of immunomodulatory, stress tolerance, disease resistant and wound-healing activities of Spirulina maxima derived pectin nanoparticles.Major revisions should be made, and the manuscript should be completed and/or modified as follows:

Reviewer comments: 1, 2, 4, 5, 6, 9, 10, 11, 12, 13, 14, 15, 16, 17, 18, 19, 21, 22, 24, 25, 26, 28, 29, 30, 31.

Author’s response:

Relevant sentences (related to above comments) were rephrased, and corrected.

Reviewer comment # 03: The authors are advised to present relevant keywords (lines 30-31).

Author’s response:

The relevant key words are included.

Reviewer comment # 07: The authors should not write Spirulina using Italic style (line 46), since it is not the Latin name of one species.

Author’s response:

It was corrected accordingly.

Reviewer comment # 08: The authors should remove some comma from sentences (ex. line 46, 54, 64, 327 etc)

Author’s response:

It was corrected accordingly.

Reviewer comment # 20: The authors should better explain the affirmation (lines 334-335): Additionally, it could be assumed that the efficacy is solely comes from the S. maxima pectin by itself.”

Author’s response:

Previously, our research group have reported the polypotency of the S. maxima pectin. In this study, we further wanted to investigate the effect of Spilulina pectin at nano scale. Here, we formed nano-scale of SmP by mechanically (sonication), rather than using any chemical methods. Thus, we assume that the whole effect is solely inherent from the pectin. We modified the sentence accordingly.  

Reviewer comment # 23: The authors should check the concentrations (line 353): 25 and 50 μg/mg.”, since previously they presented 50 μg/mL

Author’s response:

It was corrected accordingly.

Reviewer comment # 27: The authors are advised to correct the reference (line 374).

Author’s response:

It was corrected accordingly.

Reviewer comment # 32: In the M & M section, the authors should present details on preparation of SmPNP

Author’s response:

The details on SmPNPs preparation were included in the materials and methods section.

Reviewer comment # 33: The authors should also perform the evaluation of chemical composition of SmPNPs, in order to explain the obtained results. As they presented, pectins are natural compounds with heterogeneous structures, therefore the establishment of their chemical composition is essential for evaluation of biological activities

Author’s response:

Previously, our research group has determined the galacturonic acid content in the Spirulina pectins (Kyoung et al., 2020).

Reference: Kyoung, L.; Woon, Y.C.; Gun, H.P.; Younsik, J.; Areumi, P.; Yeonji, L.; Kang, D.H. Studies on the Extraction of Marine Pectin from 14 Marine Algae, Its Content and Antioxidant Activity. Korean J. Food & Nutr. 2020, 49, 677-685.

Reviewer comment # 34: The authors should better explain the novelty of their study, since some results are presented [11] subsections 4.3, 4.5, 4.6, 4.7.

Author’s response:

Main novelty of this study is that nano scale SmP is less toxic and has multifunctional activities.

Eg. Compared to SmP (Edirisinghe et al., 2019), SmPNPs was less toxic to zebrafish larvae (SmP- LD50: 330 μg/mL and SmPNPs - LD50: 547 μg/mL).

Reference: Edirisinghe, S.L.; Dananjaya, S.H.S.; Nikapitiya, C.; Liyanage, T.D.; Lee, K. A.; Oh, C.; Do-Hyung, K.; De Zoysa, M. Novel pectin isolated from Spirulina maxima enhances the disease resistance and immune responses in zebrafish against Edwardsiella piscicida and Aeromonas hydrophilaFish Shellfish Immunol, 2019, 94, 558-565.10.1016/j.fsi.2019.09.054.

Reviewer comment # 35: The authors should correct the reference (line 506).

Author’s response:

It was corrected accordingly.

Reviewer comment # 36: The authors should explain what vehicle was used (line 593) and its concentration, as well as how they choose the doses (line 596).

Author’s response:

I was explained in the text. Safe dose of SmPNPs (600 µg/fish) was optimized after a preliminary study.

Reviewer comment # 37: The authors should present references for sections 4.10 and 4.11.

Author’s response:

References were included.

Reviewer comment # 38: The authors should carefully check and correct the references, and present them in accordance with Instructions for authors (no. 3, 11, 32, etc)

Author’s response:

References were checked and modified according to the journal format.

Reviewer 2 Report

The manuscript « Spirulina maxima Derived Pectin Nanoparticles Enhance the Immunomodulation, Stress Tolerance and Wound Healing in Zebrafish” by D.C. Rajapaksha, S.L. Edirisinghe, Chamilani Nikapitiya, Hyo-Jung Kwun, Chulhong Oh, Do-Hyung Kang and Mahanama De Zoysa reports the effect of pectin nanoparticles on cultured macrophages and zebrafish (embryos and adults).

General comment:

The article is written in very bad English and should be seriously and professionally edited

The figures should be more self-explanatory.

While the authors had already published similar analysis of pectin solutions, in the present article they first sonicate the samples, describe the presence of nanoparticles and perform new experiments. We have no indications on the presence or absence of nanoparticles in the sample before sonication.

The authors should have a clear phrasing of what they are talking about in terms of expression, steady state level, protein, RNA, scavenging…(see below)

Speficic issues:

Introduction:

Line 82: “scavenging effect of SmpNPs in larvae upon the A. hydrophila infection” scavenging? See below, my comment about section 2.3.

Results:

Line 90: Why to the authors state “with irregular particle shape”. How have the authors observed this?

Line 92 – 94: “cel viability was steeply reduced at >1 mg/mL (data not shown). IC50 of the SmPNPs was 5.60 mg/mL, thus, 0.5 and 1 mg/mL of SmPNPs were selected for further in vitro experiments (data not shown)”. I consider that the authors should show these data that important to understand why this 1mg/mL dose being used.

Lines 106 – 117 & Figure 1: “Transcriptional profiling of immune-related genes” “expression of immune related genes” “mRNA expression”; the authors should be cautious that they are in fact measuring steady state mRNA levels. The authors should clearly explain the color code (green vs red / lower vs higher mRNA levels). The figure displays data that are not even mentioned in the text. The authors should not display data that are not discussed in the text. Figure 4: Please explain *, **, ***; some results have three stars (***). What does this mean?

Section 2.3: The authors report modifications of ROS levels, and through out of the manuscript discuss this in terms of scavenging (from the abstract to the discussion) or “ROS detoxification”. Do the authors have indications that this is scavenging or detoxification rather than lower production.

The authors do not report having studied H2DCFDA in SmPNPs treated fish in the absence of bacteria. Have they considered the possibility that SmPNPs could quench H2DCFDA fluorescence?

The legend of figure 2 reports “A-1, A-1’, A-2, A-2’, A-3, A-3’ and A-4, A-4’”. What do this refer to?

Figure 3: A panel, while the legend just states SmPNPs, the figure displays 25 and 50 µg. I consider that both informations ( SmPNPs and concentration) should be present in both. The figure should be more self-explanatory. The A panel should be labelled “”larvae” while the right panel “adults”

Section 2.7: According to the figures, WHR and WHP differences between treated and untreated fish are not significant. I therefore do not understand that, in the main text, the same results are described as significant.

Section 2.8: The analysis of “wound healing by histological assessment” lack some sort of quantitative analysis.

The authors states that “SmPNPs treated fish showed rapid re-epithelialization compared to the vehicle treated at 2 and 7 dpw”. Figure 8 shows no evidence of re-epithelialization for SmPNPs treated fish at 2dpw.

“The well-forming granulation of tissues was first visible in SmPNPs treated group at 2 dpw, whereas, vehicle treated group developed at 7 dpw including more inflammatory cells infiltration.”. This could be due to the poor English of the text, or this is not obvious in Figure 8.

Section 2.9: It is not clear why the authors choose kidney and muscle for their transcriptional analysis.

The authors have to be aware that that are measuring steady state mRNA levels.

Figure 9: Please explain *, **, ***

Discussion:

“Generation of oxidative stress is an imbalance of the homeostatic of redox reactions and as a result, it develops and increase the ROS [22]. Pathological effects and ROS have been demonstrated to result in infection, inflammation or subsequent cellular apoptosis”. I do not agree with the authors, ROS production is a signal being used by the organism to signal wounds or infections; it is not the cause of infections!

Line 387-: “Scavenging”, please see my comments about section 2.3

Line 415: “alph”. In the Results section, the Material and Methods section,  the supp data and in the figures, the authors report results about levels of “alp”, while in the Discussion section, they discuss “alph”. Is this the same mRNA/protein? If yes, please homogenize; if not, please clarify.

Line 433-434: “Furthermore, compared to the vehicle treated group, WHR was increased considerably in the SmPNPs treated group at 7, 10, and 14 dpw”. The WHR results reported in figure 6 (6c) are reported as “non-significant” (overlapping SD).

Lines 434-437: “Relationship between SmPNPs treatment and pigment restoration during wound healing also confirmed and SmPNPs treated fish had more re-stored pigment spots compared to the untreated fish along the whole wound healing process until 24 dpw”

The corresponding results are displayed as non-significant in figure 7B! Please clarify !

Material and methods:

Line 513: “Intercellular ROS generation”. The authors have no idea whether the detected ROS are intracellular or not; it is rather “whole body”. It is not a matter of generation, it is just a matter of quantity.

Line 514: Please provide more details relative to the imaging: wave length of excitation, wave length of imaging, reference of the camera.

Line 559: While introducing RNA preps, the authors refer to section 4.4; that is not relevant!

Line 562: The reverse transcriptions were OligodT or randomly primed?

Author Response

Reviewer #2:

Reviewer general comment: The article is written in very bad English and should be seriously and professionally edited. The figures should be more self-explanatory. While the authors had already published similar analysis of pectin solutions, in the present article they first sonicate the samples, describe the presence of nanoparticles and perform new experiments. We have no indications on the presence or absence of nanoparticles in the sample before sonication. The authors should have a clear phrasing of what they are talking about in terms of expression, steady state level, protein, RNA, scavenging.

Author’s response:

English edition was done and the certificate is attached below. We included the morphology, particle size and other physicochemical characterization results of SmPNPs as a supplementary figure 1.

Reviewer comment # 01: Introduction: Line 82: “scavenging effect of SmPNPs in larvae upon the A. hydrophila infection” scavenging? See below, my comment about section 2.3.

Author’s response:

We corrected and modified all the sections which mentioned “scavenging effect” as a “reduction of ROS”.

Reviewer comment # 02: Results: Line 90: Why to the authors state “with irregular particle shape”. How have the authors observed this?

Author’s response:

FE-SEM results showed that the SmPNPs are irregular shape. The morphology and physiochemical properties (particle size and zeta potential) of SmPNPs were included as a supplementary data.

Reviewer comment # 03: Results: Line 92 – 94: “cell viability was steeply reduced at >1 mg/mL (data not shown). IC50 of the SmPNPs was 5.60 mg/mL, thus, 0.5 and 1 mg/mL of SmPNPs were selected for further in vitro experiments (data not shown)”. I consider that the authors should show these data that important to understand why this 1mg/mL dose being used.

Author’s response:

Relevant data was included as a supplementary data.

Reviewer comment # 04: Results: Lines 106 – 117 & Figure 1: “Transcriptional profiling of immune-related genes” “expression of immune related genes” “mRNA expression”; the authors should be cautious that they are in fact measuring steady state mRNA levels. The authors should clearly explain the color code (green vs red / lower vs higher mRNA levels). The figure displays data that are not even mentioned in the text. The authors should not display data that are not discussed in the text. Figure 4: Please explain *, **, ***; some results have three stars (***). What does this mean?

Author’s response:

In the heat maps, we displayed mRNA levels of the selected genes. We corrected as “mRNA expression”or transcriptional responses. Heatmap color codes were re-organized. Asterisk marks (***) indicate statistical significance and it was described accordingly.

Reviewer comment # 05: Results: Section 2.3: The authors report modifications of ROS levels, and through out of the manuscript discuss this in terms of scavenging (from the abstract to the discussion) or “ROS detoxification”. Do the authors have indications that this is scavenging or detoxification rather than lower production. The authors do not report having studied H2DCFDA in SmPNPs treated fish in the absence of bacteria. Have they considered the possibility that SmPNPs could quench H2DCFDA fluorescence?

 Author’s response:

We corrected all the sections which mentioned “scavenging effect” as a “reduction of ROS”.The possibility of SmPNPs to quench H2DCFDA fluorescence. However, we did not optimize it. In our study, we focused to determine the effect of SmPNPs on bacteria induced ROS reduction only. In figure 2.B there is no SmPNPs concentration dependent ROS reduction. Therefore, we discussed it in the discussion that may be due to the SmPNPs could quench ROS due to its nano size.

 Reviewer comment # 06 : Results: The legend of figure 2 reports “A-1, A-1’, A-2, A-2’, A-3, A-3’ and A-4, A-4’”. What do this refer to?

Author’s response:

Related description was included.

Reviewer comment # 07: Results: Figure 3: A panel, while the legend just states SmPNPs, the figure displays 25 and 50 µg. I consider that both information’s (SmPNPs and concentration) should be present in both. The figure should be more self-explanatory. The A panel should be labelled larvae” while the right panel “adults”

 Author’s response:

It was modified accordingly.

Reviewer comment 08: Results: Section 2.7: According to the figures, WHR and WHP differences between treated and untreated fish are not significant. I therefore do not understand that, in the main text, the same results are described as significant.

 Author’s response:

It was corrected in the text (Figure 6B at 10 dpw effect was significant).

 Reviewer comment # 09: Results: Section 2.8: The analysis of “wound healing by histological assessment” lack some sort of quantitative analysis. The authors states that “SmPNPs treated fish showed rapid re-epithelialization compared to the vehicle treated at 2 and 7 dpw”. Figure 8 shows no evidence of re-epithelialization for SmPNPs treated fish at 2dpw. “The well-forming granulation of tissues was first visible in SmPNPs treated group at 2 dpw, whereas, vehicle treated group developed at 7 dpw including more inflammatory cells infiltration.”. This could be due to the poor English of the text, or this is not obvious in Figure 8.

 Author’s response:

We don’t have data for the quantitative analysis. Therefore, we showed histology figures to point out changes of wound healing upon SmPNPs treatment. As instructed, we modified the sentences related to the “re-epithelialization”.

 Reviewer comment # 10: Results: Section 2.9: It is not clear why the authors choose kidney and muscle for their transcriptional analysis. The authors have to be aware that that are measuring steady state mRNA levels.

Figure 9: Please explain *, **, ***

 Author’s response:

Kidney is the key immune related organ in fish, and muscle is the wounded site, thus, we selected these two tissues to find out the mRNA levels of immune modulatory and wound healing related gene as previously reported. The relative fold values were analyzed and fold values according to the Livek method. Asterisk marks (***) indicate statistical significance and it was described accordingly.

Ref. Livak, K.J.; Schmittgen, T.D. Analysis of relative gene expression data using real-time quantitative PCR and the 2−ΔΔCT method. Methods. 2001, 25, 402-408. 10.1006/meth.2001.1262.

 Reviewer comment # 11: Discussion: “Generation of oxidative stress is an imbalance of the homeostatic of redox reactions and as a result, it develops and increase the ROS. Pathological effects and ROS have been demonstrated to result in infection, inflammation or subsequent cellular apoptosis”. I do not agree with the authors, ROS production is a signal being used by the organism to signal wounds or infections; it is not the cause of infections!

Author’s response:

Section was corrected accordingly.

Reviewer comment # 12: Discussion: Line 387-: “Scavenging”, please see my comments about section 2.3

Author’s response:

It was corrected accordingly.

Reviewer comment # 13: Discussion: Line 415: “alph”. In the Results section, the Material and Methods section, the supp data and in the figures, the authors report results about levels of “alp”, while in the Discussion section, they discuss “alph”. Is this the same mRNA/protein? If yes, please homogenize; if not, please clarify.

Author’s response:

“alph” in the discussion was corrected as “alp”.

Reviewer comment # 14: Discussion: Line 433-434: “Furthermore, compared to the vehicle treated group, WHR was increased considerably in the SmPNPs treated group at 7, 10, and 14 dpw”. The WHR results reported in figure 6 (6c) are reported as “non-significant” (overlapping SD).

 Author’s response:

WHR difference is non-significant. We corrected the sentence according to the figure 6C.

 Reviewer comment # 15: Discussion: Lines 434-437: “Relationship between SmPNPs treatment and pigment restoration during wound healing also confirmed and SmPNPs treated fish had more re-stored pigment spots compared to the untreated fish along the whole wound healing process until 24 dpw”.The corresponding results are displayed as non-significant in figure 7B! Please clarify !

Author’s response:

Figure and the relevant text were correct.

Reviewer comment # 16: Material and methods: Line 513: “Intercellular ROS generation”. The authors have no idea whether the detected ROS are intracellular or not; it is rather “whole body”. It is not a matter of generation, it is just a matter of quantity.

 Author’s response:

It was corrected accordingly.

Reviewer comment # 17: Material and methods: Line 514: Please provide more details relative to the imaging: wave length of excitation, wave length of imaging, reference of the camera.

Author’s response:

Relevant information of camera and wavelength were included.

 Reviewer comment # 18: Material and methods: Line 559: While introducing RNA preps, the authors refer to section 4.4; that is not relevant!

 Author’s response:

It was corrected accordingly.

Reviewer comment # 19: Material and methods: Line 562: The reverse transcriptions were OligodT or randomly primed?

Author’s response:

Oligo dT was used.

Round 2

Reviewer 1 Report

The authors made the required corrections and the manuscript has significantly improved. 

Author Response

Manuscript ID: marinedrugs-937751

Title: Spirulina maxima Derived Pectin Nanoparticles Enhance the Immunomodulation, Stress Tolerance and Wound Healing in Zebrafish

Reviewers' comments:

The authors made the required corrections and the manuscript has significantly improved.

Authors response:

Although no specific comments were given, we have improved the MS by correcting and modifying the content.

Reviewer 2 Report

While the authors have addressed most of the issues raised in my first review, I consider that they have not addressed the two major issues:

  • How different are the extracts used in this study as compared to those used in their previous studies? How different are the S. maxima derived pectin preparations before and after sonication (SmPs versus SmPNP)?
  • When dealing with wound healing, they have to develop a quantitative index

Please see details below.

For this reason, I consider this article as not suitable for publication in Marine Drugs.

  • My major general concern was “We have no indications on the presence or absence of nanoparticles in the sample before sonication”. The authors have not addressed this issue!

Due to the fact that the present article mainly differs from their previous articles by the fact that sonicating their biological preparations generates nanoparticules, it is not acceptable that they do not extensively study the differences before and after sonication.

The authors should also investigate the robustness of the S. maxima derived pectin (SmP) they buy. Do all preparations display the same particles in terms of size and structure?

The images now provided in the new sup data fig1, should have been compared and discussed in the light of the images provided in their previous article dealing with S. maxima derived pectin (SmP): their references 13 and 14 in the present manuscript that also include FE-SEM images.

Reviewer comment # 02: Results: Line 90: Why to the authors state “with irregular particle shape”. How have the authors observed this?

Author’s response:

FE-SEM results showed that the SmPNPs are irregular shape. The morphology and physiochemical properties (particle size and zeta potential) of SmPNPs were included as a supplementary data.

It is good that the authors display the analysis.

The authors do not discuss the possibility that the shapes they observe are artefacts resulting from the preparation/fixation method they have used for these analyses. They should have provided details on how they prepare their samples before SEM and TEM. Have they investigated other strategies for sample preparations (deep freezing, high pressure freezing, different fixatives, environmental SEM,…)?

Reviewer comment # 09: Section 2.8: The analysis of “wound healing by histological assessment” lack some sort of quantitative analysis.

I cannot accept that the author do not set an index that would allow such a quantitative analysis.

Author Response

Manuscript ID: marinedrugs-937751

Title: Spirulina maxima Derived Pectin Nanoparticles Enhance the Immunomodulation, Stress Tolerance and Wound Healing in Zebrafish

We are very thankful to the editor-in-chief and reviewers for their valuable effort in reviewing our manuscript and giving suggestions to improve our manuscript. We have completed the 2nd revision as suggested by reviewers. Thus, we believe that current format of the manuscript can meet the journal’s criteria for publication.

Reviewers' comments:

Reviewer #2: General comments

While the authors have addressed most of the issues raised in my first review, I consider that they have not addressed the two major issues:

(a) How different are the extracts used in this study as compared to those used in their previous studies? How different are the S. maxima derived pectin preparations before and after sonication (SmPs versus SmPNP)?

(b) When dealing with wound healing, they have to develop a quantitative index.

Property

SmP

SmPNPs

Size

202 nm

125.3 nm

Zeta potential

-29.2 mv

-24.1 mv

Shape

irregular

irregular

Aggregation

Aggregated

Not aggregated

Solubility

Moderately soluble in water

Soluble in water

Author’s response:

  1. Differences of the SmPNPs (this study) was compared with SmP (previous studies) and discussed in the text (3. Discussion section) as shown below.
  2.  

In our previous studies, we discussed detailed study about the immunomodulation and wound healing properties of the SmP. According to our knowledge it was the first study which uncovered both wound healing and immune modulation properties of Spirulina maxima pectin (SmP). Therefore, considering the structural features of pectin and its effect on immune system (Popov et al, 2013) we decided to do further size reduction and produce the pectin nanomaterials, to combine the advantage of both pectin and nanoscale particles. Since, that they show great potential in some other fields (Zhao et al, 2016) and increase some physiochemical properties of marine pectin as further expansion of our previous study. Development of pectin nanoparticles through mechanical homogenization for enhancement of the properties widely applied in the research industry to get the sole effect of the pectin without any chemical incorporation (Thankappan et al, 2020 and Burapapadh et al, 2016). According to literature, pectin macromolecule contains fragments of linear and branched regions of polysaccharides such as homogalacturonan, rhamnogalacturonan-I, xylogalacturonan, and apiogalacturonan. These structural features determine the functional effect of pectin on the immune system. The fine structure of the galactan, arabinan, and apiogalacturonan side chains determines the stimulating interaction between pectin and immune cells (Popov et al, 2013). Therefore, we assumed, through mechanical homogenization such structural changes might be affected to the improved activity of SmPNPs over SmP. However, as further expansion of our study, in future we are planning to perform a more detailed study on structural changes which made to SmPNPs compared to SmP.

Reference:

  1. Popov, S. V.; Ovodov, Y. S. Polypotency of the immunomodulatory effect of pectins. Biochem. (Mosc.), 2013, 78, 823-835.
  2. Zhao, X. J.; Zhou, Z. Q. Synthesis and applications of pectin-based nanomaterials. Curr. Nanosci. 2016, 12, 103-109.
  3. Thankappan, D. A.; Raman, H. K.; Jose, J.; Sudhakaran, S. Plant-mediated biosynthesis of zein–pectin nanoparticle: Preparation, characterization, and in vitro drug release study. J King Saud Univ Sci., 2020, 32, 1785-1791.
  4. Burapapadh, K.; Takeuchi, H.; Sriamornsak, P. Development of pectin nanoparticles through mechanical homogenization for dissolution enhancement of itraconazole. Asian J. Pharm. Sci., 11, 365-375.

 (b) As suggested, we develop a quantitative index and included it as a supplementary figure 4.

Reviewer comment 01

My major general concern was “We have no indications on the presence or absence of nanoparticles in the sample before sonication”. The authors have not addressed this issue. Due to the fact that the present article mainly differs from their previous articles by the fact that sonicating their biological preparations generates nanoparticles, it is not acceptable that they do not extensively study the differences before and after sonication.

Author’s response:

SmP also can be considered as nano size material but in this study, we have more reduced the size of particles and it may alter some structural (branching, size, etc), physiochemical (solubility, aggregation, etc.) and functional properties. Therefore, we have reduced the size further and tested the biological activities.

Reviewer comment 02

The authors should also investigate the robustness of the S. maxima derived pectin (SmP) they buy. Do all preparations display the same particles in terms of size and structure?

Author’s response:

No, we did not investigate the robustness in every batch.

Reviewer comment 03

The images now provided in the new sup data fig1, should have been compared and discussed in the light of the images provided in their previous article dealing with S. maxima derived pectin (SmP): their references 13 and 14 in the present manuscript that also include FE-SEM images.

Author’s response:

The reference 13 included the FE-SEM images which showing the shape of SmP with some aggregated state. But SmPNPs showing the dispersed particles with irregular shape (Sup data fig1). As suggested, we compared and discussed the new sup fig 1 with SmP in reference 13.

Reviewer comment 04

The authors do not discuss the possibility that the shapes they observe are artefacts resulting from the preparation/fixation method they have used for these analyses.

(a). They should have provided details on how they prepare their samples before SEM and TEM. (b). Have they investigated other strategies for sample preparations (deep freezing, high pressure freezing, different fixatives, environmental SEM,)?

Author’s response:

(a). As suggested, we have provided the details of sample preparation (for SEM and TEM) in materials and methods section with relevant references.

(b). No. We did not investigate the other strategies for sample preparation.

Reviewer comment 05

I cannot accept that the author do not set an index that would allow such a quantitative analysis.

 Author’s response:

Our histology result was not very strong to perform quantitative analysis with statistical significant differences which lead us to show only the qualitative results. However, as suggested, we quantitatively analyzed the histological data and included this figure as Sup. Fig. 4.

Sup. Fig. 4. Quantitative analysis for major histological parameters in SmPNPs wound healing. A: representative images used for semi-quantitative analysis, B: quantitative analyzed criteria for neoepithelial thickness in healing wounds. C: graphs showing the nonepithelial thickness. The values are the mean ± standard deviation (SD) (n = 3). Unpaired two-tailed t-test was performed to find statistical significance (*P < .05). D: Selected histological indexes demonstrating the diminishing of abundance of inflammatory cells (inf), scab (sc), necrosis and slough (nc), re-epithelialization (ep) and formation of granulation tissue (g) on wound surface at 2 and 7 dpw. (Notes: +, slight; ++, moderate; +++, marked; -, absence).
